# Intraoperative Nerve Monitoring Parameters and Risk of Recurrent Laryngeal Nerve Injury in Thyroidectomy: A Systematic Review and Meta-Analysis

**DOI:** 10.3390/biomedicines13102516

**Published:** 2025-10-15

**Authors:** Shlomo Merchavy, Kenan Kassem, Rifat Awawde, Uday Abd Elhadi, Alaa Safia

**Affiliations:** 1Head & Neck Surgery Unit, Department of Otolaryngology, Ziv Medical Center, Safed 13100, Israel; shlomo.m@ziv.health.gov.il (S.M.);; 2Department of Internal Medicine, Emek Medical Center, Afula 18317, Israel

**Keywords:** recurrent laryngeal nerve injury, intraoperative nerve monitoring, thyroidectomy, neuromonitoring, vocal cord palsy, meta-analysis

## Abstract

**Background/Objectives:** Recurrent laryngeal nerve injury (RLNI) is a major complication of thyroidectomy, affecting voice, airway protection, and quality of life. Intraoperative nerve monitoring (IONM) has been introduced to complement direct nerve visualization and reduce RLNI risk, but its efficacy remains controversial. This systematic review and meta-analysis aimed to determine RLNI prevalence with IONM, compare rates with historical no-IONM cohorts, perform head-to-head comparisons, and assess the influence of IONM characteristics. **Methods:** PubMed, Scopus, Web of Science, Cochrane Library, and Google Scholar were searched for studies reporting RLNI rates in thyroidectomy with and without IONM. Pooled prevalence estimates were calculated for transient and permanent unilateral and bilateral RLNI in IONM studies and historical controls. Direct meta-analysis estimated pooled odds ratios (ORs) for RLNI risk reduction. Subgroup analyses examined IONM type, monitoring model, stimulation amplitude, voltage, and neuromuscular blockade use; meta-regression identified influential parameters. **Results:** A total of 103 studies involving 132,212 patients met the criteria. Unilateral transient RLNI was lower with IONM (4%, 95% CI: 4–5%) than in historical controls (5%, 95% CI: 4–6%), while unilateral permanent RLNI was 1% in both groups. Bilateral RLNI was rare. Direct comparison showed a 38% reduction in transient unilateral RLNI (OR: 0.62, 95% CI: 0.42–0.79) and a 51% reduction in permanent unilateral RLNI (OR: 0.49, 95% CI: 0.34–0.70) with IONM. Continuous IONM, lower stimulation amplitudes (≤2 mA), and avoidance of neuromuscular blockade were protective. **Conclusions:** IONM significantly reduces RLNI risk, particularly for unilateral injuries, with optimal protection achieved through continuous monitoring, low stimulation amplitudes, and avoidance of neuromuscular blockade.

## 1. Introduction

Recurrent laryngeal nerve injury (RLNI) remains one of the most concerning complications of thyroid surgery, with implications for voice function, swallowing, and, in severe cases, airway compromise [1]. While meticulous surgical technique and direct nerve visualization are considered the standard of care for RLN preservation, intraoperative nerve monitoring (IONM) has been introduced as an adjunctive tool to reduce the risk of nerve injury [2,3,4]. Despite its increasing adoption, the efficacy of IONM in preventing RLNI remains a subject of ongoing debate.

Several systematic reviews and meta-analyses [2,4,5,6,7,8,9,10,11,12] have attempted to address this question, but their findings have been inconsistent. Some studies have reported that IONM is associated with a significant reduction in transient RLNI, particularly in high-risk cases such as bilateral thyroidectomy or reoperative thyroid surgery, while others have suggested that its benefits are marginal or nonexistent compared to direct visualization alone. For instance, Pisanu et al. [4] concluded that IONM did not significantly reduce overall RLNI rates when compared to nerve visualization, while Bergenfelz et al. [6] and Ku et al. [9] found that continuous IONM was associated with lower rates of permanent RLNI. Additionally, Bai and Chen [5] demonstrated that IONM significantly reduced both transient and permanent RLNI in high-risk cases, while Higgins et al. [2] found no significant difference between IONM and direct nerve visualization alone. These conflicting results underscore the need for a more comprehensive analysis that accounts for variations in IONM techniques, monitoring settings, and patient subgroups.

A key limitation of prior meta-analyses is their failure to systematically explore the impact of specific IONM parameters on RLNI outcomes. Most existing reviews have treated IONM as a homogeneous intervention, without differentiating between continuous and intermittent IONM, varying stimulation amplitudes, or the role of neuromuscular blockade in modulating IONM effectiveness. Additionally, prior studies have largely relied on pooled prevalence estimates, whereas direct head-to-head comparisons between IONM and no IONM within the same study cohorts have been underexplored.

This systematic review and meta-analysis aim to address these gaps by providing the most comprehensive synthesis of available evidence to date. First, this study quantifies the pooled prevalence of RLNI across studies that utilized IONM and compares these rates to a historical no-IONM control group. Second, it performs a direct head-to-head comparison in studies that reported outcomes for both IONM and no IONM within the same cohort, allowing for a more precise estimation of the risk reduction associated with IONM. Third, this study systematically investigates the impact of specific IONM parameters, including the type of IONM (continuous vs. intermittent), stimulation amplitude, voltage, neuromuscular blockade use, and IONM model, to determine how these factors influence RLNI risk. Finally, meta-regression analysis is employed to quantify the relative contribution of these variables and to identify optimal conditions for IONM utilization. By incorporating a broader dataset, applying rigorous subgroup analyses, and utilizing meta-regression to refine the findings, the findings of this research have the potential to not only clarify the efficacy of IONM but also to inform surgical best practices by identifying specific conditions under which IONM provides the greatest benefit.

## 2. Materials and Methods

### 2.1. Design and Literature Search

The study protocol was registered on PROSPERO (CRD42024556259). This post-hoc systematic review and meta-analysis followed the PRISMA [13] (Preferred Reporting Items for Systematic Reviews and Meta-Analyses) and AMSTAR (Assessing the methodological quality of systematic reviews) guidelines (Appendix A) [14]. The literature search was conducted across PubMed, Scopus, Web of Science, Cochrane Library, and Google Scholar (first 200 records) up to 17 July 2024. The search strategy, detailed in Appendix A, was adapted for each database. References of included studies and related articles on PubMed and Google software were manually screened [15]. No restrictions were applied regarding the language of publication.

### 2.2. Selection Strategy

Studies were selected according to the PICOS framework [16] with the following inclusion criteria:Population: Patients undergoing thyroidectomy.Intervention: Thyroidectomy performed with intraoperative nerve monitoring (IONM), irrespective of approach or extent.Comparison: Studies without IONM (historical control group) or direct head-to-head comparisons between IONM and non-IONM groups.Outcome: The rates of unilateral and bilateral transient/permanent RLNI.Study Design: All original observational or experimental studies with >20 cases.

The exclusion criteria were:Non-original research.Abstract-only publications.Case reports or case series with <20 cases.Duplicated records or studies with overlapping datasets.Studies combining thyroid and parathyroid surgeries without stratified data for thyroidectomy.Studies not reporting RLNI outcomes.Studies reporting RLN invasion by thyroid cancer at baseline.Studies not reporting whether or not IONM was used.Animal studies.Studies focusing on irrelevant outcomes (e.g., interventional, electromyographic, or diagnostic accuracy studies).

### 2.3. Data Collection and Outcomes

A structured data extraction sheet was created in Microsoft Excel and iteratively refined to accommodate extracted data. The final sheet consisted of four sections: study characteristics, patient and surgical data, outcome data, and methodological quality. Study characteristics included authors’ names, year of publication, country, study design, sample size, and follow-up period. Patient and surgical data included age, gender, and IONM details (type, model, amplitude, voltage, neuromuscular blockade use). Outcome data included the rates of unilateral and bilateral transient/permanent RLNI. The final part covered the methodological quality assessment.

Studies published in non-English languages (Croatian, German, Iranian) were translated as needed.

### 2.4. Risk of Bias Assessment

Randomized controlled trials (RCTs) were assessed using the revised Cochrane RoB-2 tool, while observational studies were evaluated with the Newcastle–Ottawa Scale (NOS). For NOS, studies were classified according to the AHRQ thresholds: good quality = 3–4 stars in selection and 1–2 stars in comparability and 2–3 stars in outcome/exposure; fair quality = 2 stars in selection and 1–2 stars in comparability and 2–3 stars in outcome/exposure; poor quality = 0–1 star in selection or 0 stars in comparability or 0–1 star in outcome/exposure. Study selection, data extraction, and quality assessment were performed independently by 2 reviewers, with conflicts revised and resolved by the senior author.

### 2.5. Statistical Analysis

All analyses were performed using STATA (Version 18, StataCorp, College station, TX, USA) following the predefined analysis plan. Pooled RLNI rates (unilateral, bilateral, transient, permanent) were calculated for studies using IONM, both overall and by subgroup (IONM type, IONM model, amplitude, voltage, and neuromuscular blockade use). These rates were then compared to a historical no-IONM group pooled from studies that did not utilize IONM. Although the overall follow-up durations reported by studies ranged from 0.06 to 75 months, this span often reflected other outcomes (e.g., reoperation, completion thyroidectomy). For RLNI, studies uniformly categorized outcomes as transient (typically resolving within 3–12 months) or permanent (persisting beyond 12–24 months). We therefore analyzed RLNI strictly as transient vs. permanent, consistent with clinical convention.

For studies providing direct head-to-head comparisons, risk estimates for RLNI (unilateral, bilateral, transient, permanent) were calculated using random-effects meta-analysis [17]. Subgroup analyses were conducted based on IONM type, model, amplitude, voltage, and neuromuscular blockade use. Meta-regression was performed to assess the impact of these factors on RLNI rates.

Heterogeneity was assessed using the I2 statistic, with significant heterogeneity defined as I2 > 40% [18]. Sensitivity analyses included Galbraith plots to identify outliers, and publication bias was examined using funnel plots and asymmetry tests. Meta-regression models adjusted for study-level covariates, assessing multicollinearity via variance inflation factors (VIF > 5 indicated problematic multicollinearity) [19]. The reference group for categorical covariates was chosen based on the most frequently reported subgroup. A minimum of 10 studies was required for subgroup and meta-regression analyses, provided significant heterogeneity was present [20]. Consistent with our a priori threshold (≥10 studies per model), RCT-only meta-analyses/meta-regressions were not performed due to fewer than 10 eligible trials and incomplete outcome reporting. Model fit was evaluated using adjusted R-squared, with higher values indicating better fit.

Certainty of evidence was assessed with GRADE for the direct head-to-head risk comparisons. We did not apply GRADE to the single-arm pooled prevalence analyses, as no validated GRADE extension currently exists for such designs.

## 3. Results

### 3.1. Literature Search Results

The literature search and screening process yielded 4966 citations, with 2151 duplicates identified using EndNote (Figure 1). After removing duplicates, 2815 articles remained, from which 2422 were excluded during title/abstract screening. We could not retrieve the full text for 51 articles, leaving 343 for full-text review. The corresponding/first authors of those papers were contacted three times through emails and ResearchGate; however, no response was received from them. A total of 144 articles were excluded during the full-text screening phase. In summary, the reasons included no report of IONM use (n = 96), followed by no reporting of RLNI (n = 52), protocols (n = 16), review articles (n = 14), EMG studies (n = 13), irrelevant outcome data (n = 9), and abstract-only publications (n = 9). The manual search of 16,514 articles yielded 155 papers, of which 152 were screened with no additional articles being identified. Finally, 103 studies were deemed eligible for data synthesis [6,21,22,23,24,25,26,27,28,29,30,31,32,33,34,35,36,37,38,39,40,41,42,43,44,45,46,47,48,49,50,51,52,53,54,55,56,57,58,59,60,61,62,63,64,65,66,67,68,69,70,71,72,73,74,75,76,77,78,79,80,81,82,83,84,85,86,87,88,89,90,91,92,93,94,95,96,97,98,99,100,101,102,103,104,105,106,107,108,109,110,111,112,113,114,115,116,117,118,119,120,121,122].

### 3.2. Baseline Characteristics of Included Studies

The characteristics of included studies are summarized in Table 1. Most evidence was observational, with 78 retrospective cohorts, 13 prospective cohorts, and 1 cross-sectional study. Meanwhile, 11 RCTs were included. The United States accounted for the most investigated country (15 studies), followed by China (14 studies), France (8 studies), Italy (8 studies), and Turkey (8 studies), respectively. A total of 132,212 patients undergoing thyroidectomy were examined. In the 102,035 patients whom gender was disclosed, the majority were females (79,860 patients, 78.27%). The definition criteria of transient and permanent RLNI are provided in Appendix A.

### 3.3. Methodological Quality of Included Studies

The summary of the methodological quality of included observational studies is provided in Table 2. Out of 92 studies, 44 (47.83%) had good quality; 43 (46.74%) had fair quality; and 5 (5.43%) had poor quality. Out of the 11 included RCTs, eight trials had low risk of bias while the remaining three had some concerns, mainly due to lack of an in-priori protocol to assess selective reporting.

### 3.4. Pooled RLNI Rates in IONM and Historical No-IONM Groups

The pooled prevalence of recurrent laryngeal nerve injury (RLNI) was assessed in both IONM-utilizing studies and historical cohorts where no intraoperative nerve monitoring was used (Table 3). Among 87 studies reporting unilateral transient RLNI, the pooled prevalence was 4% (95% CI: 4–5%) in the IONM group, which was lower than the 5% (95% CI: 4–6%) observed in the 61 studies within the historical cohort. Similarly, for unilateral permanent RLNI, data from 54 IONM studies yielded a pooled prevalence of 1% (95% CI: 1–1%), mirroring the findings from the 39 studies in the historical group.

For bilateral transient RLNI, the pooled prevalence in 11 IONM studies was 0% (95% CI: 0–0%), which was comparable to the 0% (95% CI: 0–0%) observed in 11 historical studies. Similarly, bilateral permanent RLNI was extremely rare, with 3 IONM studies reporting a 0% (95% CI: 0–0%) prevalence, comparable to 4 historical studies, which also showed 0% (95% CI: 0–0%).

### 3.5. Subgroup Analysis of RLNI Rates Based on IONM Characteristics

Subgroup analysis was conducted to determine how different IONM parameters influence RLNI prevalence (Table 3). When comparing IONM type, the prevalence of unilateral transient RLNI remained relatively stable across studies using continuous IONM (42 studies; 4%, 95% CI: 3–6%), intermittent IONM (16 studies; 5%, 95% CI: 3–6%), and those where the IONM type was not reported (17 studies; 5%, 95% CI: 3–6%). However, intermittent IONM appeared to be associated with a slightly lower prevalence of unilateral permanent RLNI (8 studies; 0%, 95% CI: 0–1%), compared to 1% (28 studies; 95% CI: 0–1%) in continuous IONM.

Analysis based on IONM model revealed notable variation. The AVALANCHE system had the highest pooled prevalence for unilateral transient RLNI (2 studies; 6%, 95% CI: 1–10%), while the CLEO nerve monitor (1 study) and Neurosign System (3 studies) exhibited some of the lowest rates (1–2%). For bilateral transient RLNI, the Medtronic NIM 3.0 system showed an increased prevalence (1 study; 4%, 95% CI: 0–9%) compared to other models.

When examining stimulation amplitude, lower amplitudes (<1 mA) were associated with an increased prevalence of unilateral transient RLNI (3 studies; 7%, 95% CI: 5–9%), whereas 1 mA stimulation showed a reduced rate (15 studies; 3%, 95% CI: 2–4%). A similar trend was noted for neuromuscular blockade use, where studies that explicitly did not use neuromuscular blockade reported a lower prevalence of unilateral transient RLNI (26 studies; 3%, 95% CI: 3–4%), compared to those using neuromuscular blockade (17 studies; 7%, 95% CI: 2–12%).

Lastly, stratification by stimulation voltage demonstrated that the 100 μV threshold was associated with a higher pooled prevalence of unilateral transient RLNI (14 studies; 5%, 95% CI: 3–6%), while a more moderate prevalence (3–4%) was observed with higher voltage thresholds (4 studies).

### 3.6. Direct Head-to-Head Comparison Between IONM and No IONM: Unilateral Transient RLNI

A direct comparison between studies that reported outcomes for both IONM and no IONM demonstrated a significant protective effect of IONM in reducing the risk of unilateral transient RLNI (Figure 2). The pooled OR for unilateral transient RLNI was 0.62 (95% CI: 0.42–0.79, *p* < 0.001, very low certainty). Despite the presence of substantial heterogeneity (I^2^ = 79.91%), the leave-one-out sensitivity analysis showed that results remained consistent across iterations (Appendix A). The risk of publication bias was insignificant (Egger’s *p* = 0.2373) (Appendix A).

Further subgroup analyses explored the influence of various IONM parameters on the protective effect (Figure 3). The reduction in RLNI risk was evident across different IONM types, with continuous IONM associated with an OR of 0.61 (95% CI: 0.44–0.83), while intermittent IONM exhibited a slightly attenuated effect (OR = 0.72, 95% CI: 0.42–1.22), though the latter did not reach statistical significance. When stratified by IONM model, the Medtronic NIM 2.0 system demonstrated the most pronounced protective effect, with an OR of 0.42 (95% CI: 0.24–0.73, *p* = 0.002). Similarly, the Medtronic NIM 3.0 system showed a borderline significant reduction in RLNI risk (OR = 0.55, 95% CI: 0.32–0.94, *p* = 0.028). The impact of stimulation amplitude was also examined, revealing that higher stimulation amplitudes (>3 mA) were associated with an increased RLNI risk, whereas amplitudes of ≤2 mA exhibited stronger protective effects. Additionally, the use of neuromuscular blockade appeared to diminish the efficacy of IONM, as evidenced by a higher OR of 1.36 (95% CI: 0.79–2.56, *p* = 0.076) in cases where neuromuscular blockade was used, suggesting that avoiding neuromuscular blockade may enhance the protective effect of IONM.

### 3.7. Direct Head-to-Head Comparison Between IONM and No IONM: Unilateral Permanent RLNI

A direct comparison between IONM and no IONM demonstrated a significant protective effect of IONM in reducing the risk of unilateral permanent RLNI (Figure 4). The pooled OR for unilateral permanent RLNI was 0.49 (95% CI: 0.34–0.70, *p* < 0.001, low certainty). Heterogeneity was low (I^2^ = 17.07%), and the leave-one-out sensitivity analysis showed no remarkable change in reported estimate (Appendix A). Publication bias was insignificant (Egger’s *p* = 0.5417) (Appendix A).

Further subgroup analysis was performed to examine the impact of different IONM parameters on the protective effect (Figure 5). The benefit of IONM was observed across various subgroups, with continuous IONM showing an OR of 0.61 (95% CI: 0.44–0.86), while intermittent IONM exhibited an even lower OR of 0.35 (95% CI: 0.14–0.89). Among IONM models, Medtronic NIM 2.0 provided the greatest risk reduction (OR = 0.41, 95% CI: 0.16–1.04), followed by the Medtronic Xomed 2.0 system (OR = 0.55, 95% CI: 0.16–1.93). However, subgroup comparisons did not yield statistically significant differences (*p*-values > 0.05), suggesting that the protective effect of IONM was largely consistent across different models and settings. Stimulation amplitude analysis revealed that lower amplitudes (≤2 mA) were associated with a greater protective effect, while amplitudes > 3 mA exhibited weaker risk reduction. Additionally, voltage did not appear to significantly modify the effect size.

### 3.8. Meta-Regression Analysis for the Direct Head-to-Head Comparison Between IONM and No IONM

To further explore the determinants of RLNI risk reduction associated with IONM, a meta-regression analysis was performed (Table 4). The models assessed the impact of IONM type, IONM model, stimulation amplitude, and neuromuscular blockade use on the risk of transient unilateral and bilateral RLNI. Due to multicollinearity, voltage was excluded from the final models.

For transient unilateral RLNI, continuous IONM was associated with a significantly lower risk compared to intermittent IONM (β = −1.196, *p* = 0.045), reinforcing the notion that real-time monitoring may provide superior nerve protection. Among IONM models, the Medtronic NIM 2.0 system exhibited the greatest reduction in RLNI risk (β = −1.931, *p* = 0.016), while the Medtronic Xomed 2.0 model was associated with an increased risk (β = 2.643, *p* = 0.004). The Medtronic (version not specified) system also showed a significant reduction in RLNI risk (β = −1.099, *p* = 0.045), whereas the Neurosign System, Inomed System, and CLEO nerve monitor did not demonstrate statistically significant effects. Stimulation amplitude, modeled as a continuous variable per mA increase, did not significantly influence the risk of RLNI (β = 0.925, *p* = 0.228). Similarly, the use of neuromuscular blockade was not significantly associated with RLNI risk in this model (β = −0.528, *p* = 0.172). The overall model fit indicated no residual heterogeneity (R^2^ = 100%; I^2^ = 0%), suggesting that the included predictors explained all the observed variability.

For transient bilateral RLNI, none of the included covariates reached statistical significance. Continuous IONM did not show a significant advantage over intermittent IONM (β = 1.015, *p* = 0.691). Among IONM models, neither the Medtronic NIM 2.0, Medtronic Xomed 2.0, nor the Neurosign System demonstrated a statistically significant effect. The model fit again suggested that all variability was explained by the included predictors (R^2^ = 100%; I^2^ = 0%), though the lack of significant findings indicates that the determinants of transient bilateral RLNI may be more complex or influenced by unmeasured factors.

## 4. Discussion

RLNI remains a significant complication of thyroid surgery, with serious implications for voice function, airway protection, and overall patient quality of life. While IONM has been widely adopted to mitigate this risk, its efficacy has remained a subject of debate. The findings of this systematic review and meta-analysis provide strong evidence supporting the protective role of IONM, particularly in reducing the incidence of transient and permanent unilateral RLNI. Additionally, this study highlights key determinants that influence the effectiveness of IONM, underscoring the importance of optimizing monitoring parameters to maximize its clinical benefit.

### 4.1. Comparison with Prior Evidence

The conclusions of this study build upon and refine the findings of previous meta-analyses that have assessed the impact of IONM on RLNI. Zheng et al. [12] conducted a meta-analysis incorporating over 36,000 nerves at risk and demonstrated a statistically significant reduction in total RLNI with IONM, with an odds ratio of 0.74 (95% CI: 0.59–0.92), particularly for transient injuries. Similarly, Rulli et al. [11] reported that IONM was associated with a reduction in transient RLNI, with a relative risk of 0.73 (95% CI: 0.54–0.98, *p* = 0.035), but found no significant effect on permanent RLNI. The present study aligns with these findings but extends the analysis further by incorporating extensive subgroup analyses and meta-regression, revealing the influence of IONM type, stimulation parameters, and neuromuscular blockade on RLNI outcomes.

While some prior meta-analyses have questioned the overall benefit of IONM, particularly with respect to permanent RLNI, others have suggested that the protective effects of IONM are most pronounced in high-risk surgical settings, such as bilateral thyroidectomies or oncologic resections. Bergenfelz et al. [6] found that although IONM did not significantly reduce the overall incidence of early RLNI, it was associated with a lower risk of permanent vocal cord palsy, with an odds ratio of 0.43 (95% CI: 0.19–0.93). Ku et al. [9] demonstrated that continuous IONM was particularly effective, reporting a permanent RLNI rate of only 0.05%. These findings are supported by the results of the present study, which confirm that continuous IONM provides superior nerve protection compared to intermittent IONM.

Conversely, other studies have reported conflicting findings. Pisanu et al. [4] and Higgins et al. [2] failed to identify a significant difference in RLNI rates when comparing IONM with direct nerve visualization alone. These discrepancies may stem from differences in surgical expertise, patient selection criteria, and variations in the application of IONM protocols. The meta-regression analysis in this study addresses this gap by identifying specific factors—such as the choice of IONM model, the applied stimulation amplitude, and the avoidance of neuromuscular blockade—that significantly influence RLNI risk reduction.

### 4.2. Key Contributions and Novel Insights

One of the primary strengths of this study lies in its comprehensive approach, integrating pooled prevalence estimates with direct head-to-head comparisons. By synthesizing data from over 130,000 patients, this study provides one of the most extensive analyses to date, offering high-powered evidence in favor of IONM. The findings demonstrate a 38% reduction in the odds of transient unilateral RLNI (OR: 0.62, 95% CI: 0.42–0.79, *p* < 0.001) and a 51% reduction in the odds of permanent unilateral RLNI (OR: 0.49, 95% CI: 0.34–0.70, *p* < 0.001). These results reinforce the role of IONM as a critical adjunct in thyroidectomy.

Beyond demonstrating the overall benefit of IONM, the findings of this study highlight the importance of optimization in monitoring parameters. Continuous IONM was associated with greater nerve protection compared to intermittent IONM. The Medtronic NIM 2.0 system exhibited the most substantial risk reduction, while the Medtronic Xomed 2.0 model was paradoxically associated with an increased risk of RLNI. Lower stimulation amplitudes (≤2 mA) were found to be more protective than higher amplitudes (>3 mA). Additionally, the use of neuromuscular blockade was found to diminish the efficacy of IONM, reinforcing the need for careful anesthetic management to ensure optimal monitoring performance. The meta-regression analysis quantitatively validated these findings, demonstrating that variations in IONM technique significantly influence RLNI risk.

While these subgroup and meta-regression findings help highlight potentially modifiable monitoring parameters, they should be interpreted with appropriate caution where contributing study counts are small and confidence intervals widen. In such cases, results are best viewed as hypothesis-generating signals that warrant confirmation in adequately powered prospective studies.

The geographic distribution of included studies (spanning Europe, Asia, and the Americas) is also noteworthy. Differences in surgical training, adoption of IONM technology, and perioperative standards across regions may have contributed to heterogeneity in reported RLNI outcomes. For example, European and East Asian centers often report higher adoption of continuous IONM, whereas many North American series rely on intermittent IONM. Such regional variation highlights the need for future multinational prospective studies with standardized protocols to ensure broader applicability of results.

### 4.3. Clinical Implications

The findings of this study have significant implications for surgical practice. While IONM has already been widely adopted in thyroid surgery, the focus should shift from merely using IONM to ensuring that it is applied in a standardized and optimized manner. The results suggest that rather than a binary decision regarding whether to use IONM, surgeons should focus on how IONM is implemented. Specifically, the data support the use of continuous IONM with optimized stimulation settings and the avoidance of neuromuscular blockade to maximize nerve protection.

These findings also have important implications for high-risk thyroidectomy cases. While the absolute risk of bilateral RLNI remains low, the results indicate that IONM reduces this risk, reinforcing its role in complex cases such as total thyroidectomy, re-operative thyroid surgery, and malignancy-related thyroid resections. In these scenarios, where the stakes of nerve injury are particularly high, the use of IONM may play a pivotal role in improving surgical safety.

### 4.4. Limitations and Future Directions

Despite its strengths, this study is not without limitations. One of the primary concerns is the heterogeneity among the included studies, particularly regarding differences in surgical technique, IONM protocols, and follow-up durations. We limited “no-IONM” prevalence comparators to cohorts that explicitly reported no IONM use; nonetheless, such contrasts can still be confounded by secular trends in technique, case selection, and perioperative care. Accordingly, we treat these contrasts as contextual rather than causal, and center our conclusions on the head-to-head analyses. Another limitation is the heterogeneity in how transient and permanent RLNI were defined across studies (Appendix A), which prevented consistent categorization and precluded sensitivity analyses restricted to standardized definitions; this variability may have contributed to the observed heterogeneity in pooled estimates.

While the meta-regression analysis attempts to account for these variables, residual confounding remains possible. Additionally, publication bias is an inherent challenge in meta-analyses. Although the funnel plot analysis did not detect significant bias, the potential for selective reporting cannot be entirely excluded. Study-level covariates requested by the reviewer (year, RCT vs. cohort, surgeon/center volume) were reported too sparsely and inconsistently—especially within the <10 RCT subset—to support stable multivariable meta-regression; we therefore prioritized the prespecified IONM-parameter models and transparently report this limitation.

Despite pre-specified safeguards (e.g., conducting subgroup/meta-regression only when ≥10 studies were available and heterogeneity was present) and performing sensitivity checks, several subgroups still included a limited number of studies. Small subgroup samples increase imprecision, widen confidence intervals, and can heighten the probability of spurious or unstable estimates—particularly in the context of multiple comparisons. Accordingly, subgroup and meta-regression effects should be interpreted as exploratory and hypothesis-generating rather than definitive, pending confirmation in standardized, prospective datasets. Although all covariates were pre-specified in the protocol, we limited the presented analyses to IONM-related parameters; nonetheless, the performance of multiple subgroup and regression models may increase the risk of inflated type I error, and findings should therefore be interpreted with caution.

Future research should prioritize prospective, standardized studies with uniform IONM protocols, particularly in the form of randomized controlled trials comparing continuous versus intermittent IONM. Further investigation into machine-learning-assisted IONM interpretation may also offer new opportunities for real-time risk stratification and intraoperative decision-making. By refining these parameters, future studies can help further clarify the optimal use of IONM in thyroid surgery.

## 5. Conclusions

This systematic review and meta-analysis provide compelling evidence that IONM significantly reduces the risk of both transient and permanent RLNI. The findings clarify that the effectiveness of IONM is contingent upon how it is implemented rather than its mere presence or absence. Standardization of IONM protocols, including the use of continuous monitoring, appropriate stimulation parameters, and avoidance of neuromuscular blockade, is essential to maximize its protective effect. These results support the routine adoption of IONM in thyroidectomy, emphasizing the need for a structured and evidence-based approach to ensure optimal surgical outcomes. Future research should focus on refining these findings with high-quality prospective trials to further establish best practices in RLNI prevention.

## Figures and Tables

**Figure 1 biomedicines-13-02516-f001:**
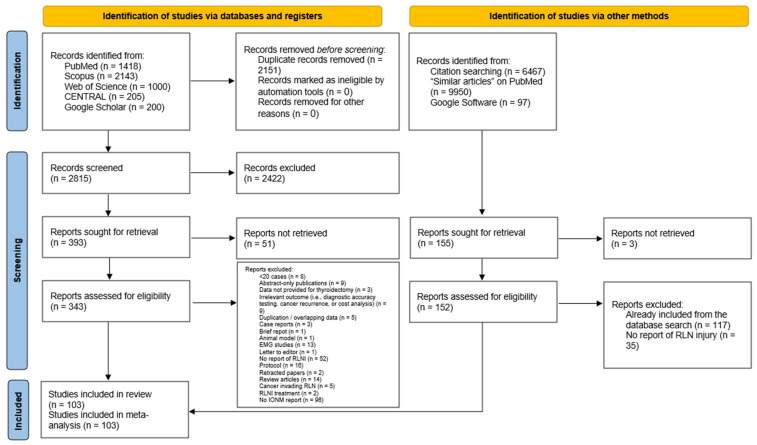
PRISMA flow diagram showing the results of the database search and screening processes.

**Figure 2 biomedicines-13-02516-f002:**
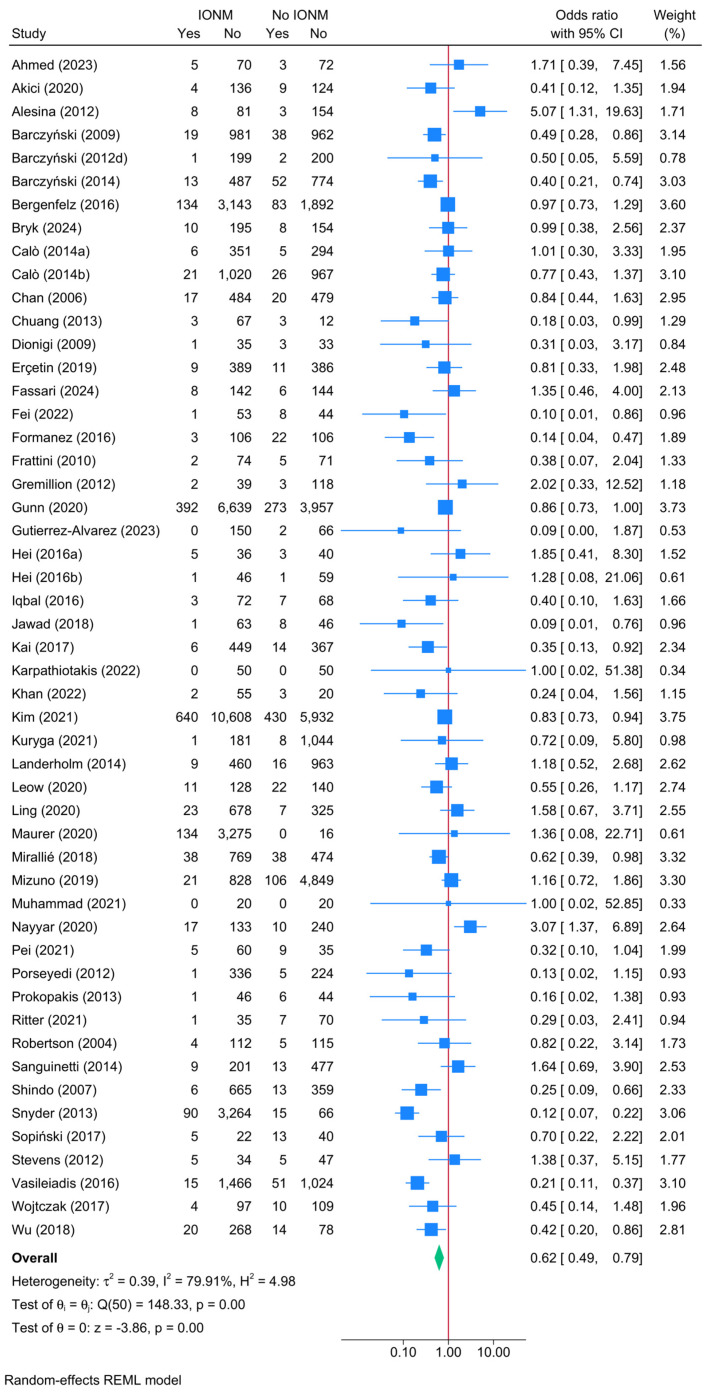
Forest plot showing the risk of unilateral transient RLNI in studies reporting direct head-to-head comparison between IONM and no IONM [6,23,24,26,33,36,37,41,42,43,44,48,50,52,54,55,56,57,58,59,61,63,64,66,67,70,71,72,73,74,76,77,84,86,87,90,92,94,96,97,100,101,104,108,110,111,112,114,116,118,119,120,121,122].

**Figure 3 biomedicines-13-02516-f003:**
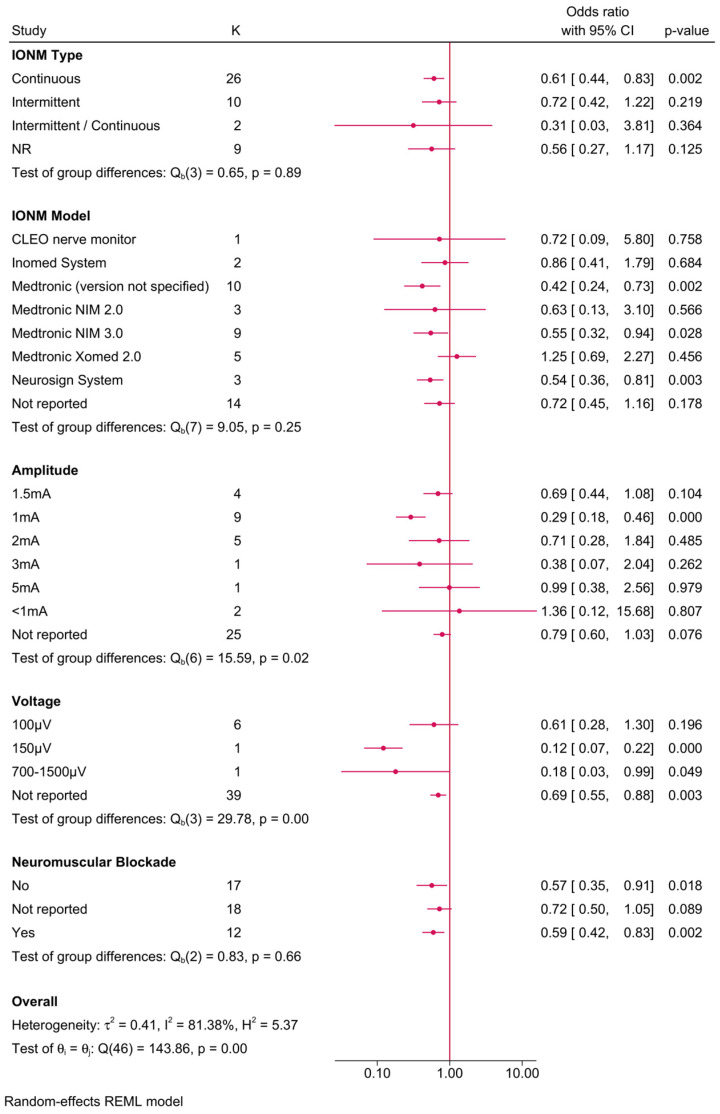
Forest plot showing the risk of unilateral transient RLNI in studies reporting direct head-to-head comparison between IONM and no IONM, stratified by IONM-related parameters (type, model, amplitude/voltage, and neuromuscular blockade use).

**Figure 4 biomedicines-13-02516-f004:**
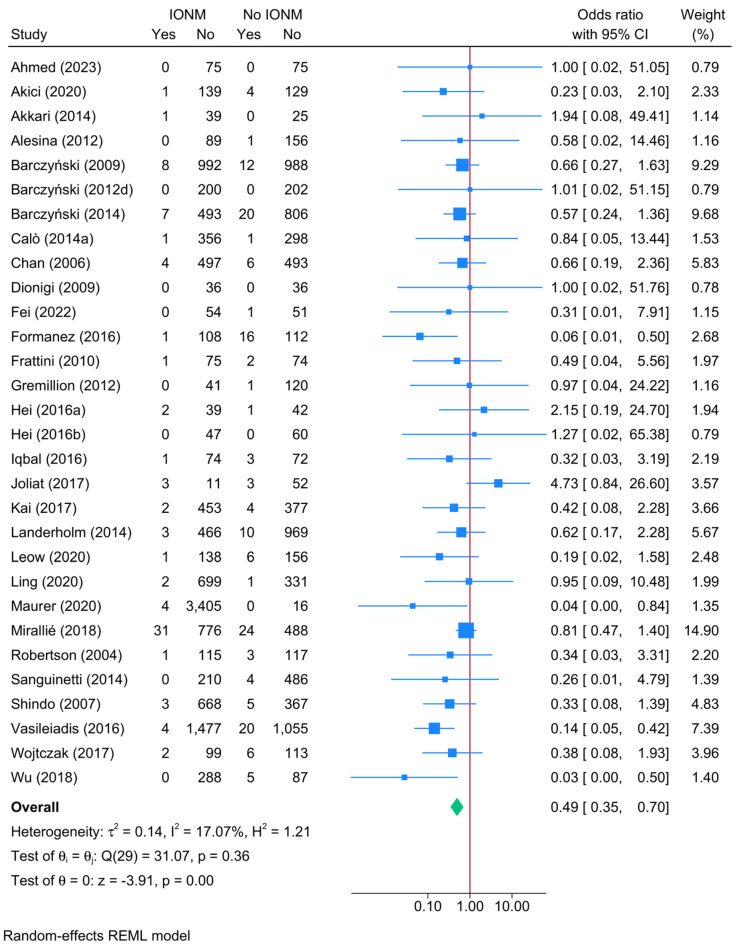
Forest plot showing the risk of unilateral permanent RLNI in studies reporting direct head-to-head comparison between IONM and no-IONM [23,24,25,26,33,36,37,43,44,50,55,56,57,58,63,64,66,68,70,74,76,77,84,86,101,104,108,114,116,118].

**Figure 5 biomedicines-13-02516-f005:**
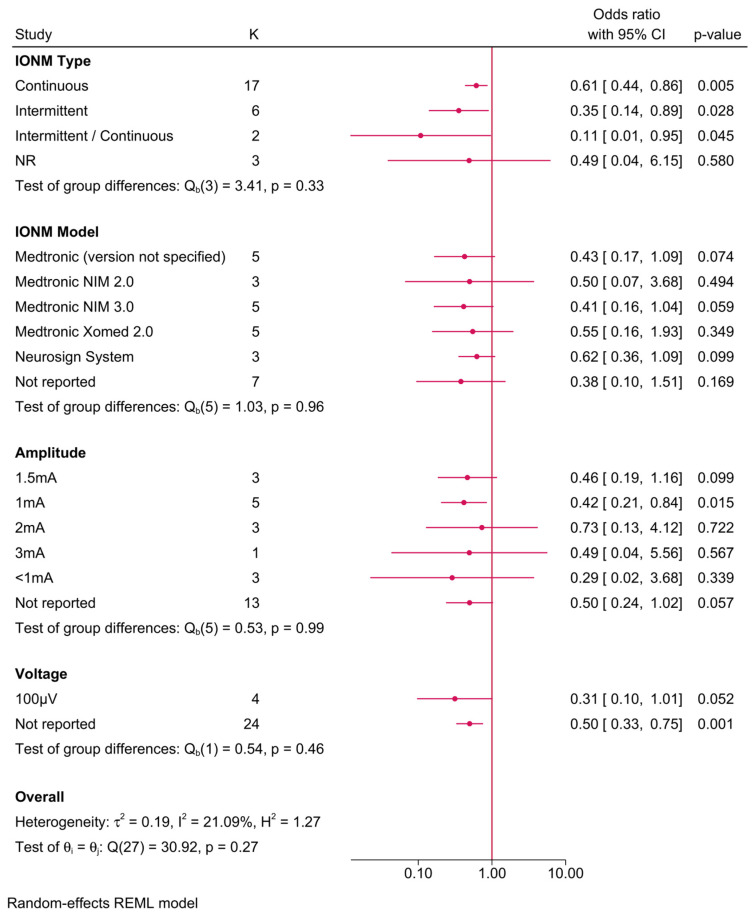
Forest plot showing the risk of unilateral permanent RLNI in studies reporting direct head-to-head comparison between IONM and no IONM, stratified by IONM-related parameters (type, model, amplitude/voltage, and neuromuscular blockade use).

**Table 1 biomedicines-13-02516-t001:** Baseline characteristics of studies reporting the use of IONM during thyroid surgery and reporting RLNI as an outcome.

Author (YOP)	Country	Design	YOI	Sample	Age		Gender		FU (mo)	Neuromonitoring (IONM)					
					Mean	SD	M	F		Yes/No	Type	Model	Amplitude	Voltage	Neuromuscular Blockade
Acun (2004a) [21]	Turkey	RCT	2001–2003	152	43	(24–77)	39	113	12	0/152	-	-	-	-	-
Acun (2005) [22]	Turkey	RC	-	176	44	(23–77)	49	127	-	0/176	-	-	-	-	-
Ahmed (2023) [23]	Iraq	RCT	2018–2020	150	39	-	22	128	-	75/75	Continuous	Not reported	Not reported	Not reported	No
Akici (2020) [24]	Turkey	RC	2012–2017	273	47	12	38	235	-	140/133	Continuous	Not reported	1.5 mA	Not reported	No
Akkari (2014) [25]	France	RC	2004–2012	65	12.5	0.7	16	49	-	40/25	Continuous	Medtronic Xomed 2.0	<1 mA	Not reported	No
Alesina (2012) [26]	Germany	RC	1999–2011	246	55	12.5	37	209	-	89/157	Continuous	Medtronic Xomed 2.0	<1 mA	Not reported	No
Al-Hakami (2019) [27]	KSA	RC	2008–2017	456	42.6	(10–89)	99	357	12	456/0	-	Not reported	Not reported	Not reported	No
Alhan (2015) [28]	Turkey	RC	2004–2012	620	48	14	109	511	6	0/620	-	-	-	-	-
Alharbi (2018) [29]	KSA	RC	2011–2018	320	42.25	9.5	112	208	-	0/620	-	Not reported	-	-	No
Alqahtani (2023) [30]	KSA	RC	2015–2021	432	41.2	19.1	76	361	0.75	0/432	-	Not reported	-	-	No
Ambe (2014) [31]	Germany	RC	2006–2012	305	54.3	14.25	78	227	-	305/0	-	Not reported	Not reported	Not reported	No
Aygun (2022) [32]	Turkey	RC	2016–2021	871	49.17	13.42	199	672	-	871/0	Intermittent	Medtronic Xomed 2.0	1 mA	100 μV	Yes
											Continuous	Medtronic (version not specified)	1 mA	500 μV	Yes
Barczyński (2009) [33]	Poland	RCT	2006–2007	1000	51.6	14.6	88	912	12	500/500	Continuous	Neurosign System	1 mA	Not reported	Yes
Barczyński (2010) [34]	Poland	RCT	2000–2003	600	47.22	15.61	53	517	-	0/600	-	Not reported	-	-	Yes
Barczyński (2012c) [35]	Poland	RCT	2000–2004	191	45.9	(43.1–48.9)	21	170	12	0/191	-	-	-	-	-
Barczyński (2012d) [36]	Poland	RCT	2009–2010	210	49.9	14.7	0	201	6	100/101	Continuous	Medtronic NIM 3.0	1 mA	Not reported	Yes
Barczyński (2014) [37]	Poland	RC	1993–2012	854	54.3	13.4	687	167	-	306/548	Continuous	Neurosign System	1 mA	Not reported	Yes
Bawa (2021) [38]	KSA	RC	2013–2019	339	38	(29–48)	59	280	-	0/339	-	Not reported	-	-	Not reported
Bergenfelz (2016) [6]	Sweden	RC	2009–2013	5252	49	(38–63)	1050	4202	6	3277/1975	-	Not reported	Not reported	Not reported	Not reported
Bertelli (2021) [39]	Brazil	RC	2017	93	-	-	14	79	-	93/0	-	Neurosoft (INTRO)	Not reported	Not reported	Not reported
Bihain (2021) [40]	France	PC	2013–2019	603	52.8	15	137	466	0.06	367/236	Continuous	Medtronic NIM 3.0	1 mA	100 μV	Yes
Bryk (2024) [41]	Poland	RC	-	367	52.2	(18–79)	55	312	-	205/162	Continuous	Inomed System	5 mA	Not reported	No
Calò (2014a) [42]	Italy	RC	2007–2013	656	-	-	-	-	-	357/299	Continuous	Medtronic Xomed 2.0	Not reported	Not reported	No
Calò (2014b) [43]	Italy	RC	2007–2012	2034	-	-	-	-	-	1041/993	Continuous	Medtronic Xomed 2.0	Not reported	Not reported	No
Chan (2006) [44]	China	RC	2002–2005	639	49	(8–93)	133	506	-	501/499	Continuous	Neurosign System	1.5 mA	Not reported	No
Chen (2022a) [45]	China	RC	2019–2020	110	41.1	7.25	46	64	-	110/0	-	Not reported	Not reported	Not reported	Not reported
Chiang (2004) [46]	Taiwan	RC	1986–2002	521	42	(17–78)	118	403	-	0/521	-	Not reported	-	-	Not reported
Chiang (2011) [47]	Taiwan	RC	2006–2009	231	-	-	-	-	-	231/0	Continuous	Medtronic Xomed 2.0	1 mA	Not reported	No
Chuang (2013) [48]	Taiwan	RC	2001–2010	71	-	(22.8–85)	12	59	-	56/15	Continuous	Medtronic (version not specified)	1 mA	700–1500 μV	No
Dedhia (2020) [49]	USA	RC	2000–2018	1096	50.1	0.78	341	755	6	1096/0	-	Not reported	Not reported	Not reported	Not reported
Dionigi (2009) [50]	Italy	RCT	2004–2007	72	40.5	(19–77)	10	62	12	-	-	Medtronic NIM 2.0	-	-	-
Dralle (2004) [51]	Germany	RC	1998–2001	16,448	-	-	-	-	-	12,166/17,832	Continuous	Neurosign System	5 mA	Not reported	No
Erçetin (2019) [52]	Turkey	PC	2008–2016	748	47.8	13	130	665	12	398/397	Intermittent	Medtronic (version not specified)	1.5 mA	Not reported	No
Farizon (2017) [53]	France	RC	2012–2015	195	53.4	(14–88)	34	161	12	195/0	Continuous	Medtronic (version not specified)	<1 mA	100 μV	No
Fassari (2024) [54]	Italy	RC	-	300	48.6	11.9	121	179	-	150/150	Intermittent	Medtronic (version not specified)	2 mA	Not reported	No
Fei (2022) [55]	China	RC	2013–2018	106	32.25	6.83	-	-	-	54/52	Intermittent/Continuous	Medtronic (version not specified)	2 mA	100 μV	Yes
Formanez (2016) [56]	Philippines	RC	2009–2014	237	41	(20–65)	74	163	6	109/128	-	Not reported	Not reported	Not reported	Not reported
Frattini (2010) [57]	Italy	RC	-	152	40.6	(19–77)	67	85	12	76/76	Continuous	Medtronic (version not specified)	3 mA	Not reported	No
Gremillion (2012) [58]	USA	RC	2007–2010	119	-	-	-	-	-	31/88	Continuous	Not reported	Not reported	Not reported	Not reported
Gunn (2020) [59]	USA	RC	2016–2017	11,370	53	(41–63)	2476	8894	1	7031/4230	Continuous	Not reported	Not reported	Not reported	Not reported
Gür (2019) [60]	Turkey	RC	2014–2017	456	52.8	(18–82)	106	350	-	456/0	Continuous	AVALANCHE	2 mA	100 μV	No
Gutierrez-Alvarez (2023) [61]	UAS	RC	2019–2022	218	-	-	39	179	-	150/68	Continuous	Medtronic NIM 3.0	Not reported	Not reported	No
Hamilton (2019) [62]	UK	RC	2014–2016	256	-	27–86	27	169	-	-	-	APS Medtronic	-	-	-
Hei (2016a) [63]	China	RCT	2012–2014	70	47.5	9.9	16	54	6	41/43	Intermittent	Medtronic NIM 2.0	Not reported	Not reported	Yes
Hei (2016b) [64]	China	RC	2009–2011	97	45.35	11.23	19	78	-	46/51	Intermittent	Medtronic NIM 2.0	2 mA	Not reported	Yes
Hu (2016) [65]	China	RC	2003–2014	5559	55	(9–87)	714	4845	6	0/5559	-	Not reported	-	-	Not reported
Iqbal (2016) [66]	Pakistan	RCT	2013–2014	150	-	13–60	53	97	-	75/75	-	Not reported	Not reported	Not reported	Not reported
Jawad (2018) [67]	Baghdad	PC	2012–2016	132	37.35	8.37	47	85	6	64/54	-	Not reported	Not reported	Not reported	Not reported
Joliat (2017) [68]	Switzerland	RC	2005–2013	451	50	43–63	12	51	12	8/55	-	Not reported	Not reported	Not reported	Not reported
Jonas (2006) [69]	France	PC	1999–2004	937	50.8	(24–83)	-	-	12	-	-	Neurosign System	-	-	-
Kai (2017) [70]	China	RC	2013–2016	522	65.66	0.3	122	430	-	340/212	Continuous	Medtronic NIM 3.0	1 mA	100 μV	Yes
Karpathiotakis (2022) [71]	Italy	PC	2018–2020	100	55	(43–65)	17	83	6	50/50	Intermittent	Medtronic NIM 3.0	Not reported	100 μV	No
Khan (2022) [122]	Pakistan	CS	2020–2021	70	44.43	-	18	52	6	57/23	-	Not reported	Not reported	Not reported	Not reported
Kim (2021) [72]	USA	RC	2016–2018	17,610	52	15	3904	13706	-	11,248/6362	-	Not reported	Not reported	Not reported	Not reported
Kuryga (2021) [73]	Poland	RC	2005–2012	1235	49.5	-	169	1065	36	182/1052	Continuous	CLEO nerve monitor	1 mA	Not reported	Yes
Landerholm (2014) [74]	Sweden	PC	1984–2011	973	54.7	16.1	242	1080	12	0/973	-	Not reported	-	-	Not reported
Lenay-Pinon (2021) [75]	France	RC	2013–2019	1026	53	(18–81)	266	760	12	-	-	Inomed System	-	-	-
Leow (2020) [76]	Singapore	RC	2014–2018	261	49.2	12.5	68	193	-	108/153	Intermittent	Medtronic NIM 3.0	Not reported	Not reported	No
Ling (2020) [77]	China	RC	2012–2017	1696	52	(40–59)	280	753	6	1104/592	Intermittent	Medtronic NIM 3.0	2 mA	100 μV	Yes
Liu (2020) [78]	China	RC	2017–2019	2350	51.9	13.3	371	1887	6	2350/0	Intermittent	Medtronic NIM 3.0	1 mA	100 μV	No
Liu (2021) [79]	China	RC	2012–2019	415	35.45	(19–48)	1	404	6	415/0	Intermittent	Medtronic NIM 3.0	1 mA	100 μV	Yes
Machens (2018) [80]	Germany	RC	1994–2017	167	6.9	-	78	89	6	167/0	Continuous	Not reported	Not reported	Not reported	Not reported
Mahoney (2021) [81]	USA	RC	2016–2017	11,552	≥65	-	2514	9038	-	7130/4422	-	Not reported	Not reported	Not reported	Not reported
Maksimoski (2022) [82]	USA	RC	2012–2017	1025	13.9	-	228	797	-	795/230	-	Not reported	Not reported	Not reported	Not reported
Marin Arteaga (2018) [83]	Switzerland	RC	2012–2016	1001	55	16.8	35	66	6	1001/0	Continuous	Medtronic NIM 3.0	Not reported	100 μV	No
Maurer (2020) [84]	Germany	RC	2017–2019	1808	44	(14–80)	330	1478	-	3409/16	Intermittent/Continuous	Not reported	Not reported	Not reported	Not reported
Messenbaeck (2018) [85]	Austria	RC	-	246	45.6	(21–73)	29	217	-	246/0	Continuous	Medtronic (version not specified)	Not reported	Not reported	Not reported
Mirallié (2018) [86]	France	PC	2012–2014	1328	51.2	(18–80)	267	1061	6	807/521	Continuous	Medtronic (version not specified)	Not reported	Not reported	Yes
Mizuno (2019) [87]	Japan	RC	2008–2017	5084	57.7	14.6	1528	4276	1	849/4955	-	Not reported	Not reported	Not reported	Not reported
Mohammad (2022) [88]	Kuwait	RC	2016–2019	197	49	23–85	71	126	6	171/26	Continuous	Medtronic NIM 3.0	2 mA	100 μV	No
Moreira (2020) [89]	Australia	RC	2010–2017	1003	-	-	220	783	-	1003/0	Continuous	Medtronic Xomed 2.0	Not reported	Not reported	Not reported
Muhammad (2021) [90]	Malaysia	RCT	2016	25	54.2	-	7	33	-	20/20	Continuous	Medtronic NIM 3.0	1 mA	Not reported	Yes
Nagaoka (2022) [91]	Japan	RC	2016–2020	100	36.2	-	1	99	6	25/75	-	Not reported	Not reported	Not reported	Not reported
Nayyar (2020) [92]	India	RC	2017–2019	228	-	-	150	250	-	150/250	-	Not reported	Not reported	Not reported	Not reported
Paek (2022) [93]	Korea	RC	2013–2014	315	42.45	9.9	70	245	6	315/0	Continuous	Medtronic NIM 3.0	Not reported	Not reported	Not reported
Pei (2021) [94]	China	RC	2010–2020	109	49.56	14.98	48	61	-	65/44	Continuous	Medtronic NIM 3.0	2 mA	Not reported	Not reported
Périé (2013) [95]	France	PC	2007–2011	100	47.1	16–81	19	81	6	-	-	Neurosign System	-	-	-
Porseyedi (2012) [96]	Iran	RC	2005–2011	566	40.26	-	124	442	-	337/229	-	Not reported	Not reported	Not reported	Not reported
Prokopakis (2013) [97]	Greece	RC	2004–2011	97	61	47–75	20	77	-	-	-	Medtronic (version not specified)	-	-	-
Raval (2009) [98]	USA	RC	2000–2007	31	12.2	(5–17)	6	25	6	23/8	Continuous	Medtronic (version not specified)	Not reported	Not reported	Not reported
Razavi (2018) [99]	USA	RC	2016–2017	27	41.3	12.2	4	23	3	-	-	Medtronic (version not specified)	-	-	-
Ritter (2021) [100]	Israel	RC	2001–2019	113	13.5	3.9	29	84	12	-	-	Medtronic NIM 2.0	-	-	-
Robertson (2004) [101]	USA	RC	1999–2002	165	44.4	-	54	182	-	82/83	Continuous	Medtronic Xomed 2.0	Not reported	Not reported	Not reported
Rudolph (2014) [102]	France	RC	1991–2006	494	39	-	41	453	6	494/0	-	Not reported	Not reported	Not reported	Not reported
Russell (2021) [103]	USA	RC	2017–2020	533	44	(10–84)	90	443	6	533/0	Continuous	Medtronic NIM 3.0	Not reported	Not reported	Not reported
Sanguinetti (2014) [104]	Italy	RC	2012	350	-	-	-	-	-	105/245	Continuous	Medtronic Xomed 2.0	Not reported	Not reported	Not reported
Sarkis (2017) [105]	Australia	RC	1990–2014	7406	-	-	-	-	3	7406/0	Continuous	Not reported	Not reported	Not reported	Not reported
Schneider (2019) [106]	Austria	PC	2012–2016	4707	-	-	1212	3495	12	4707/0	Intermittent	AVALANCHE	2 mA	Not reported	Not reported
Sena (2019) [107]	Italy	RC	2009–2018	237	52.7	-	89	199	-	-	-	Medtronic NIM 3.0	-	-	-
Shindo (2007) [108]	USA	RC	1998–2005	684	-	-	-	-	-	671/372	Continuous	Medtronic (version not specified)	Not reported	Not reported	No
Snyder (2010) [109]	USA	RC	2003–2009	1242	57.3	-	-	-	-	1242/0	-	Not reported	Not reported	Not reported	Not reported
Snyder (2013) [110]	USA	RC	2004–2011	1936	52	-	685	2750	-	3354/81	Continuous	Medtronic (version not specified)	1 mA	150 μV	No
Sopiński (2017) [111]	China	RCT	2014–2016	80	57.95	9.35	4	76	-	27/53	Intermittent	Inomed System	Not reported	Not reported	Yes
Stevens (2012) [112]	USA	PC	2004–2008	91	48.45	12.9	37	54	6	39/52	Continuous	Medtronic (version not specified)	Not reported	100 μV	Not reported
Tabriz (2024) [113]	Germany	RC	2016–2020	1147	52	(13–90)	293	854	-	1147/0	Intermittent	Not reported	Not reported	Not reported	Not reported
Vasileiadis (2016) [114]	Greece	RC	2002–2012	2566	51.35	14.18	528	2028	12	1481/1075	Intermittent	Medtronic NIM 2.0	1 mA	Not reported	No
Velayutham (2022) [115]	India	PC	2017–2019	84	-	-	-	-	-	84/0	Continuous	Medtronic NIM 3.0	2.5 mA	500 μV	No
Wojtczak (2017) [116]	Poland	PC	2011–2014	632	53.94	13.87	117	515	6	236/396	Intermittent	Medtronic NIM 3.0	1.5 mA	Not reported	Not reported
Wu (2017) [117]	Taiwan	PC	2012–2014	323	50	(16–83)	63	260	6	-	-	Medtronic NIM 3.0	-	-	-
Wu (2018) [118]	USA	RC	2006–2015	380	38.5	14.04	71	309	-	288/92	Continuous	Medtronic (version not specified)	<1 mA	100 μV	Not reported
Xu (2023) [119]	China	RC	2015–2021	416	37.8	7.87	-	-	6	416/0	Intermittent	Medtronic (version not specified)	3 mA	Not reported	Not reported
Yu (2020) [120]	China	RC	2016–2017	93	50	24–78	22	71	-	93/0	Continuous	Medtronic (version not specified)	Not reported	Not reported	Yes
Yuksekdag (2019) [121]	Turkey	RC	2014–2018	260	51	32–67	-	-	6	-	-	Medtronic NIM 3.0	-	-	-

YOP: year of publication; RC: retrospective cohort; RCT: randomized controlled trial; PC: prospective cohort; IONM: intraoperative nerve monitoring; RLNI: recurrent laryngeal nerve injury; SD: standard deviation; M: male; F: female; FU: follow-up; mo: month: NR: not reported.

**Table 2 biomedicines-13-02516-t002:** A summary of the methodological quality of included observational studies using the Newcastle–Ottawa Scale.

Author (YOP)	Selection				Comparability		Outcome			Overall Rating
	Representativeness of the Exposed Cohort	Selection of the Non-Exposed Cohort	Ascertainment of Exposure	Demonstration That Outcome of Interest Was Not Present at Start of Study	Design	Analysis	Assessment of Outcome	Was Follow-Up Long Enough for Outcomes to Occur?	Adequacy of Follow-Up of Cohorts	
Acun (2005) [21]	Yes	Yes	Yes	Yes	Yes	No	Yes	Yes	Yes	Good
Akici (2020) [24]	Yes	Yes	Yes	Yes	Yes	No	Yes	Yes	Yes	Good
Akkari (2014) [25]	No	Yes	Yes	Yes	Yes	No	Yes	No	No	Poor
Alesina (2012) [26]	Yes	Yes	Yes	Yes	Yes	No	Yes	No	No	Fair
AlHakami (2019) [27]	Yes	Yes	Yes	Yes	Yes	No	Yes	No	No	Fair
Alhan (2015) [28]	Yes	Yes	Yes	Yes	Yes	No	Yes	No	No	Fair
Alharbi (2018) [29]	Yes	Yes	Yes	Yes	Yes	No	Yes	No	No	Fair
Alqahtani (2023) [30]	Yes	Yes	Yes	Yes	Yes	No	Yes	No	No	Fair
Ambe (2014) [31]	Yes	Yes	Yes	Yes	Yes	No	Yes	No	No	Fair
Aygun (2022) [32]	Yes	Yes	Yes	Yes	Yes	No	Yes	No	No	Fair
Barczyński (2014) [33]	Yes	Yes	Yes	Yes	Yes	No	Yes	No	No	Fair
Bawa (2021) [38]	Yes	Yes	Yes	Yes	Yes	No	Yes	No	No	Fair
Bergenfelz (2016) [6]	Yes	Yes	Yes	Yes	Yes	No	Yes	No	No	Fair
Bertelli (2021) [39]	No	Yes	Yes	Yes	Yes	No	Yes	Yes	Yes	Fair
Bihain (2021) [40]	Yes	Yes	Yes	Yes	Yes	No	Yes	No	No	Fair
Bryk (2024) [41]	Yes	Yes	Yes	Yes	Yes	No	Yes	No	No	Fair
Calò (2014a) [42]	Yes	Yes	Yes	Yes	Yes	No	Yes	Yes	Yes	Good
Calò (2014b) [43]	Yes	Yes	Yes	Yes	Yes	No	Yes	No	No	Fair
Chan (2006) [44]	Yes	Yes	Yes	Yes	Yes	No	Yes	No	No	Fair
Chen (2022a) [45]	Yes	Yes	Yes	Yes	Yes	No	Yes	Yes	Yes	Good
Chiang (2004) [46]	Yes	Yes	Yes	Yes	Yes	No	Yes	No	No	Fair
Chiang (2011) [47]	Yes	Yes	Yes	Yes	Yes	No	Yes	No	No	Fair
Chuang (2013) [48]	No	Yes	Yes	Yes	Yes	No	Yes	Yes	Yes	Fair
Dedhia (2020) [49]	Yes	Yes	Yes	Yes	Yes	No	Yes	Yes	Yes	Good
Dralle (2004) [51]	Yes	Yes	Yes	Yes	Yes	No	Yes	Yes	Yes	Good
Erçetin (2019) [52]	Yes	Yes	Yes	Yes	Yes	No	Yes	Yes	Yes	Good
Farizon (2017) [53]	Yes	Yes	Yes	Yes	Yes	No	Yes	No	No	Fair
Fassari (2024) [54]	Yes	Yes	Yes	Yes	Yes	No	Yes	Yes	Yes	Good
Fei (2022) [55]	Yes	Yes	Yes	Yes	Yes	No	Yes	Yes	Yes	Good
Formanez (2016) [56]	Yes	Yes	Yes	Yes	Yes	No	Yes	No	No	Fair
Frattini (2010) [57]	Yes	Yes	Yes	Yes	Yes	No	Yes	No	No	Fair
Gremillion (2012) [58]	Yes	Yes	Yes	Yes	Yes	No	Yes	Yes	Yes	Good
Gunn (2020) [59]	Yes	Yes	Yes	Yes	Yes	No	Yes	No	No	Fair
Gür (2019) [60]	Yes	Yes	Yes	Yes	Yes	No	Yes	Yes	Yes	Good
GutierrezAlvarez (2023) [61]	Yes	Yes	Yes	Yes	Yes	No	Yes	Yes	Yes	Good
Hamilton (2019) [62]	Yes	Yes	Yes	Yes	Yes	No	Yes	Yes	Yes	Good
Hei (2016b) [63]	No	Yes	Yes	Yes	Yes	No	Yes	No	No	Poor
Hu (2016) [64]	Yes	Yes	Yes	Yes	Yes	No	Yes	Yes	Yes	Good
Jawad (2018) [67]	Yes	Yes	Yes	Yes	Yes	No	Yes	No	No	Fair
Joliat (2017) [68]	Yes	Yes	Yes	Yes	Yes	No	Yes	No	No	Fair
Jonas (2006) [69]	Yes	Yes	Yes	Yes	Yes	No	Yes	No	No	Fair
Kai (2017) [70]	Yes	Yes	Yes	Yes	Yes	No	Yes	Yes	Yes	Good
Karpathiotakis (2022) [71]	Yes	Yes	Yes	Yes	Yes	No	Yes	Yes	Yes	Good
Khan (2022) [122]	Yes	Yes	Yes	Yes	Yes	No	Yes	Yes	Yes	Good
Kim (2021) [72]	Yes	Yes	Yes	Yes	Yes	No	Yes	Yes	Yes	Good
Kuryga (2021) [73]	Yes	Yes	Yes	Yes	Yes	No	Yes	Yes	Yes	Good
Landerholm (2014) [74]	Yes	Yes	Yes	Yes	Yes	No	Yes	No	No	Fair
LenayPinon (2021) [75]	Yes	Yes	Yes	Yes	Yes	No	Yes	Yes	Yes	Good
Leow (2020) [76]	Yes	Yes	Yes	Yes	Yes	No	Yes	Yes	Yes	Good
Ling (2020) [77]	Yes	Yes	Yes	Yes	Yes	No	Yes	Yes	Yes	Good
Liu (2020) [78]	Yes	Yes	Yes	Yes	Yes	No	Yes	No	No	Fair
Liu (2021) [79]	Yes	Yes	Yes	Yes	Yes	No	Yes	Yes	Yes	Good
Machens (2018) [80]	Yes	Yes	Yes	Yes	Yes	No	Yes	Yes	Yes	Good
Mahoney (2021) [81]	Yes	Yes	Yes	Yes	Yes	No	Yes	Yes	Yes	Good
Maksimoski (2022) [82]	Yes	Yes	Yes	Yes	Yes	No	Yes	Yes	Yes	Good
Marin Arteaga (2018) [83]	Yes	Yes	Yes	Yes	Yes	No	Yes	No	No	Fair
Maurer (2020) [84]	Yes	Yes	Yes	Yes	Yes	No	Yes	No	No	Fair
Messenbaeck (2018) [85]	Yes	Yes	Yes	Yes	Yes	No	Yes	Yes	Yes	Good
Mirallié (2018) [86]	Yes	Yes	Yes	Yes	Yes	No	Yes	No	No	Fair
Mizuno (2019) [87]	Yes	Yes	Yes	Yes	Yes	No	Yes	No	No	Fair
Mohammad (2022) [88]	Yes	Yes	Yes	Yes	Yes	No	Yes	Yes	Yes	Good
Moreira (2020) [89]	Yes	Yes	Yes	Yes	Yes	No	Yes	Yes	Yes	Good
Nagaoka (2022) [91]	Yes	Yes	Yes	Yes	Yes	No	Yes	Yes	Yes	Good
Nayyar (2020) [92]	Yes	Yes	Yes	Yes	Yes	No	Yes	No	No	Fair
Paek (2022) [93]	Yes	Yes	Yes	Yes	Yes	No	Yes	No	No	Fair
Pei (2021) [94]	Yes	Yes	Yes	Yes	Yes	No	Yes	No	No	Fair
Périé (2019) [95]	Yes	Yes	Yes	Yes	Yes	No	Yes	Yes	Yes	Good
Porseyedi (2012) [96]	Yes	Yes	Yes	Yes	Yes	No	Yes	Yes	Yes	Good
Prokopakis (2013) [97]	No	Yes	Yes	Yes	Yes	No	Yes	No	No	Poor
Raval (2009) [98]	No	Yes	Yes	Yes	Yes	No	Yes	No	No	Poor
Razavi (2018) [99]	No	Yes	Yes	Yes	Yes	No	Yes	Yes	Yes	Fair
Ritter (2021) [100]	Yes	Yes	Yes	Yes	Yes	No	Yes	No	No	Fair
Robertson (2004) [101]	Yes	Yes	Yes	Yes	Yes	No	Yes	No	No	Fair
Rudolph (2014) [102]	Yes	Yes	Yes	Yes	Yes	No	Yes	No	No	Fair
Russell (2021) [103]	Yes	Yes	Yes	Yes	Yes	No	Yes	Yes	Yes	Good
Sanguinetti (2014) [104]	Yes	Yes	Yes	Yes	Yes	No	Yes	Yes	Yes	Good
Sarkis (2017) [105]	Yes	Yes	Yes	Yes	Yes	No	Yes	Yes	Yes	Good
Schneider (2019) [106]	Yes	Yes	Yes	Yes	Yes	No	Yes	Yes	Yes	Good
Sena (2019) [107]	Yes	Yes	Yes	Yes	Yes	No	Yes	Yes	Yes	Good
Shindo (2007) [108]	Yes	Yes	Yes	Yes	Yes	No	Yes	No	No	Fair
Snyder (2010) [109]	Yes	Yes	Yes	Yes	Yes	No	Yes	Yes	Yes	Good
Snyder (2013) [110]	Yes	Yes	Yes	Yes	Yes	No	Yes	Yes	Yes	Good
Stevens (2012) [112]	No	Yes	Yes	Yes	Yes	No	Yes	Yes	Yes	Fair
Tabriz (2024) [113]	Yes	Yes	Yes	Yes	Yes	No	Yes	Yes	Yes	Good
Vasileiadis (2016) [114]	Yes	Yes	Yes	Yes	Yes	No	Yes	No	No	Fair
Velayutham (2022) [115]	No	Yes	Yes	Yes	Yes	No	Yes	No	No	Poor
Wojtczak (2017) [116]	Yes	Yes	Yes	Yes	Yes	No	Yes	Yes	Yes	Good
Wu (2017) [117]	Yes	Yes	Yes	Yes	Yes	No	Yes	No	No	Fair
Wu (2018) [118]	Yes	Yes	Yes	Yes	Yes	No	Yes	Yes	Yes	Good
Xu (2023) [119]	Yes	Yes	Yes	Yes	Yes	No	Yes	Yes	Yes	Good
Yu (2020) [120]	No	Yes	Yes	Yes	Yes	No	Yes	Yes	Yes	Fair
Yuksekdag (2019) [121]	Yes	Yes	Yes	Yes	Yes	No	Yes	Yes	Yes	Good

**Table 3 biomedicines-13-02516-t003:** A summary of the pooled prevalence rate of RLNI in studies using IONM and historical cohorts (no IONM use) with subsets based on IONM details.

		Unilateral RLNI				Bilateral RLNI			
		Transient		Permanent		Transient		Permanent	
		IONM	Historical	IONM	Historical	IONM	Historical	IONM	Historical
Main-Pooled	Studies	87	61	54	39	11	11	3	4
	Patients	11,248	6362	7406	5559	4955	4955	1352	2341
	Proportion (95% CI)	4% (4–5)	5% (4–6)	1% (1–1)	1% (1–1)	0% (0–0)	0% (0–0)	0% (0–0)	0% (0–0)
IONM Type	Continuous	4% (3–6)		1% (0–1)		0% (0–1)			
	Intermittent	5% (3–6)		0% (0–1)		4% (0–9)			
	Not reported	5% (3–6)		1% (1–2)		0% (0–0)			
IONM Model	AVALANCHE	6% (1–10)		1% (1–1)		-			
	CLEO nerve monitor	1% (0–2)		-		1% (0–2)			
	Inomed System	10% (0–23)		-		-			
	Medtronic (version not specified)	7% (1–12)		1% (0–2)		1% (0–1)			
	Medtronic NIM 2.0	2% (0–5)		0% (0–1)		-			
	Medtronic NIM 3.0	4% (3–5)		0% (0–1)		4% (0–9)			
	Medtronic Xomed 2.0	4% (2–6)		0% (0–0)		0% (0–0)			
	Neurosign System	2% (2–3)		2% (0–3)		-			
	Neurosoft (INTRO)	3% (0–7)		1% (0–3)		-			
	Not reported	5% (3–6)		1% (0–1)		0% (0–0)			
Amplitude	<1 mA	7% (5–9)		0% (0–1)		1% (0–2)			
	1 mA	3% (2–4)		0% (0–1)		1% (0–2)			
	1.5 mA	3% (2–4)		1% (0–2)		-			
	2 mA	5% (3–7)		1% (0–1)		-			
	2.5 mA	5% (1–9)		-		-			
	3 mA	4% (2–6)		1% (0–4)		1% (0–2)			
	5 mA	5% (2–8)		4% (3–4)		-			
	Not reported	5% (4–7)		1% (0–1)		0% (0–0)			
Voltage	100 μV	5% (3–6)		0% (0–1)		4% (0–9)			
	150 μV	3% (2–3)		-		-			
	500 μV	4% (3–6)		-		-			
	700–1500 μV	4% (0–9)		-		-			
	Not reported	4% (3–5)		1% (1–1)		0% (0–0)			
Neuromuscular blockade	No	3% (3–4)		1% (1–2)		0% (0–0)			
	Yes	7% (2–12)		1% (0–2)		1% (0–1)			
	Not reported	5% (4–6)		0% (0–1)		0% (0–0)			

Historical cohorts refer to the studies that reported no use of IONM either before IONM introduction or for not using it after production. CI: confidence interval; IONM: intraoperative nerve monitoring; RLNI: recurrent laryngeal nerve injury.

**Table 4 biomedicines-13-02516-t004:** Meta-regression models predicting the risk of unilateral and bilateral transient RLNI based on IONM characteristics.

Transient Unilateral RLNI	Coefficient	SE	Z	*p* Value	Low CI	High CI
Continuous IONM (vs. Intermittent)	−1.196	0.597	−2.000	0.045	−2.365	−0.027
IONM Model (Reference: Medtronic NIM 3.0)						
Medtronic (version not specified)	−1.099	0.549	−2.000	0.045	−2.176	−0.022
Medtronic Xomed 2.0	2.643	0.916	2.890	0.004	0.847	4.438
Neurosign System	0.539	0.440	1.220	0.221	−0.324	1.402
Medtronic NIM 2.0	−1.931	0.801	−2.410	0.016	−3.500	−0.362
Inomed System	−2.924	3.059	−0.960	0.339	−8.919	3.071
CLEO nerve monitor	0.989	1.134	0.870	0.383	−1.233	3.211
Amplitude (per mA change)	0.925	0.768	1.200	0.228	−0.580	2.429
Neuromuscular blockade use (vs. non)	−0.528	0.387	−1.360	0.172	−1.286	0.230
Constant	−0.517	1.438	−0.360	0.719	−3.335	2.301
Model Fit	R^2^ = 100%; I^2^ = 0%					
**Transient Bilateral RLNI**	**Coefficient**	**SE**	**Z**	***p* Value**	**Low CI**	**High CI**
Continuous IONM (vs. Intermittent)	1.015	2.553	0.400	0.691	−3.989	6.019
IONM Model (Reference: Medtronic NIM 3.0)						
Medtronic (version not specified)	−1.255	2.418	−0.520	0.604	−5.995	3.484
Medtronic Xomed 2.0	1.498	1.701	0.880	0.378	−1.836	4.831
Neurosign System	0.262	0.858	0.300	0.761	−1.420	1.944
Medtronic NIM 2.0	−0.172	2.283	−0.080	0.940	−4.646	4.302
Amplitude (per mA change)	1.697	2.093	0.810	0.417	−2.405	5.799
Neuromuscular blockade use (vs. non)	−0.012	0.719	−0.020	0.986	−1.421	1.396
Constant	−3.436	4.202	−0.820	0.414	−11.672	4.800
Model Fit	R^2^ = 100%; I^2^ = 0%					

SE: standard error; CI: confidence interval; IONM: intraoperative nerve monitoring; RLNI: recurrent laryngeal nerve injury.

## Data Availability

Not applicable.

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
