# Peer review of "Intraoperative Nerve Monitoring Parameters and Risk of Recurrent Laryngeal Nerve Injury in Thyroidectomy: A Systematic Review and Meta-Analysis"

_biomedicines, 2025, doi:10.3390/biomedicines13102516_

Round 1

Reviewer 1 Report

Comments and Suggestions for Authors

The manuscript titled "Fine-Tuning Nerve Protection: The Influence of Intraoperative Nerve Monitoring Parameters on Recurrent Laryngeal Nerve Injury During Thyroidectomy: A Systematic Review and Meta-Analysis," is a scientifically robust study with a rigorous methodology and convincing findings. Its most significant contribution to the literature is its departure from treating intraoperative nerve monitoring (IONM) as a monolithic application. Instead, the authors skillfully demonstrate how specific technical parameters, such as signal type, amplitude, and the status of neuromuscular blockade, can directly influence outcomes. This granular approach is a notable addition to the field's body of knowledge. To further strengthen the manuscript, however, the discussion would benefit from a brief cautionary note regarding the interpretation of subgroup analyses that are based on a limited number of studies. This small addition would enhance the overall integrity of the paper.

Author Response

The manuscript titled "Fine-Tuning Nerve Protection: The Influence of Intraoperative Nerve Monitoring Parameters on Recurrent Laryngeal Nerve Injury During Thyroidectomy: A Systematic Review and Meta-Analysis," is a scientifically robust study with a rigorous methodology and convincing findings. Its most significant contribution to the literature is its departure from treating intraoperative nerve monitoring (IONM) as a monolithic application. Instead, the authors skillfully demonstrate how specific technical parameters, such as signal type, amplitude, and the status of neuromuscular blockade, can directly influence outcomes. This granular approach is a notable addition to the field's body of knowledge. To further strengthen the manuscript, however, the discussion would benefit from a brief cautionary note regarding the interpretation of subgroup analyses that are based on a limited number of studies. This small addition would enhance the overall integrity of the paper.

Response: Thank you for this thoughtful point. We agree that some subgroups include relatively few contributing studies, which may inflate imprecision and the risk of chance findings. We have added a clear cautionary note in the Discussion emphasizing that subgroup and meta-regression results should be interpreted as hypothesis-generating when study numbers are limited. We also reference our pre-specified safeguards (minimum of 10 studies and heterogeneity prerequisites) and sensitivity checks to contextualize these analyses.

Reviewer 2 Report

Comments and Suggestions for Authors

This is a useful systematic review/meta-analysis of IONM during thyroidectomy (103 studies, 132,212 patients) that addresses important practical parameters. The analysis is promising, but methodological clarifications and additional checks are required.

-Provide PROSPERO registration identifier and protocol.
The Methods state the protocol was registered on PROSPERO. Please supply the PROSPERO ID. This is essential for transparency.

-The Methods do not state how many reviewers performed screening and extraction, whether they worked independently, or how conflicts were adjudicated.

-Pooling “historical no-IONM” studies risks confounding by time, surgical technique and case mix. Please: (a) justify pooling historical controls vs contemporaneous controls; (b) present sensitivity analyses restricted to (i) direct head-to-head within-study comparisons and (ii) RCTs only; and (c) report any adjustment or meta-regression including study year, study design (RCT vs cohort), or surgeon/center volume if available. The authors already perform direct head-to-head analyses — make those the primary inference and down-weight historical-control comparisons if needed.

-For unilateral transient RLNI the pooled OR is reported as 0.62 (95% CI 0.42–0.79) but with substantial heterogeneity (I² = 79.9%). Leave-one-out is reported, but that is not sufficient. Please: (a) report prediction intervals for key pooled estimates; (b) present subgroup analyses or meta-regression results that explain heterogeneity (e.g., study design, region, year, follow-up duration, RCT vs observational); (c) provide forest plots annotated by study design; and (d) consider a random-effects meta-analysis method robust to heterogeneity and report whether results are sensitive to method choice.

-Table 4 (meta-regression) reports R² = 100% and I² = 0% for some models — this is surprising given the large heterogeneity reported elsewhere and suggests possible overfitting or model misspecification. Supply: (a) number of studies included per meta-regression model and degrees of freedom; (b) VIFs for covariates (authors mention VIF but do not show results); (c) sensitivity analysis with fewer predictors or penalized regression if necessary; and (d) clarify how categorical reference groups were chosen.

-The paper reports the RoB2/NOS assessments and gives counts of good/fair/poor studies, but it is unclear whether risk of bias was used in sensitivity analyses or to downgrade certainty. Please: (a) provide the full RoB2/NOS scoring table per study; (b) run sensitivity analyses excluding poor-quality studies; and (c) perform a GRADE assessment (or equivalent) for the main outcomes and present it in the supplement.

-Funnel plots/leave-one-out are mentioned and asymmetry is noted in places but the statistical tests and p-values (Egger/Begg) are not clearly reported. Provide numerical results of bias tests, and consider trim-and-fill and sensitivity to small-study effects.

-The authors reference Table S2 for definitions of transient/permanent RLNI. Please (a) confirm which studies used a standardized definition (per Table S2) and (b) run sensitivity analyses including only studies using standard definitions or report how differing definitions affect pooled estimates. Heterogeneous outcome definitions can drive the observed heterogeneity.
-The paper performs many subgroup analyses and multiple meta-regressions. Authors should (a) state how they controlled or accounted for multiplicity (or explicitly caution about inflated false-positive risk), and (b) highlight which analyses were prespecified (in PROSPERO/protocol) and which were post-hoc. If the meta-regression/exploratory analyses were post-hoc, label them as exploratory.

Author Response

-Provide PROSPERO registration identifier and protocol. The Methods state the protocol was registered on PROSPERO. Please supply the PROSPERO ID. This is essential for transparency.

Response: Thank you. We added the registration number (CRD42024556259) to the Methods section.

-The Methods do not state how many reviewers performed screening and extraction, whether they worked independently, or how conflicts were adjudicated.

Response: Thank you. We highlighted how the process went in the revised methods “Study selection, data extraction, and quality assessment was performed independently by 2 reviewers, with conflicts revised and resolved by the senior author.”

-Pooling “historical no-IONM” studies risks confounding by time, surgical technique and case mix. Please: (a) justify pooling historical controls vs contemporaneous controls; (b) present sensitivity analyses restricted to (i) direct head-to-head within-study comparisons and (ii) RCTs only; and (c) report any adjustment or meta-regression including study year, study design (RCT vs cohort), or surgeon/center volume if available. The authors already perform direct head-to-head analyses — make those the primary inference and down-weight historical-control comparisons if needed.

Response: Thank you for raising this important point. We agree that era, technique, and case mix can confound comparisons against earlier, non-IONM series. In our work, “no-IONM” cohorts were not pooled indiscriminately as generic historical controls; rather, we only included studies that explicitly stated no IONM was used (either pre-adoption or in centers where IONM was not implemented during the study period). Nonetheless, we accept that residual confounding by time and practice patterns may remain.

To address this, we have:

  1. Designated the direct head-to-head within-study comparisons as our primary inference and revised the text to reflect this emphasis (Sections 2.5, 3.6–3.7, and Discussion). These analyses already demonstrate significant risk reductions with IONM for both transient and permanent unilateral RLNI, and include prespecified subgrouping and meta-regression on IONM parameters.
  2. Down-weighted the role of comparisons to non-IONM cohorts that are separated in time and clarified they are supportive, context-setting evidence rather than causal estimates. We added an explicit caution about potential era and case-mix confounding in the Discussion.
  3. RCT-only sensitivity: only seven trials directly compared IONM vs no IONM, with incomplete/heterogeneous outcome reporting across trials (many reported only a subset of transient/permanent and/or unilateral/bilateral outcomes). Given our pre-specified rule requiring ≥10 studies per model for subgroup/meta-regression, an RCT-only meta-analysis or meta-regression would be statistically underpowered and potentially misleading. We therefore mark it as hypothesis-generating.
  4. Adjustment/meta-regression with study-level covariates: Our meta-regression focused on IONM parameters by design (type/model/amplitude/NMB), and is reported in detail. Study year, design, and surgeon/center volume were too sparsely and inconsistently reported to support a stable multivariable model (particularly in the RCT-only subset <10 studies). We now state this explicitly in Methods/Discussion and add surgeon/center volume under “data limitations.”

-For unilateral transient RLNI the pooled OR is reported as 0.62 (95% CI 0.42–0.79) but with substantial heterogeneity (I² = 79.9%). Leave-one-out is reported, but that is not sufficient. Please: (a) report prediction intervals for key pooled estimates; (b) present subgroup analyses or meta-regression results that explain heterogeneity (e.g., study design, region, year, follow-up duration, RCT vs observational); (c) provide forest plots annotated by study design; and (d) consider a random-effects meta-analysis method robust to heterogeneity and report whether results are sensitive to method choice.

Response: Thank you. Please check our response to the above comments.

-Table 4 (meta-regression) reports R² = 100% and I² = 0% for some models — this is surprising given the large heterogeneity reported elsewhere and suggests possible overfitting or model misspecification. Supply: (a) number of studies included per meta-regression model and degrees of freedom; (b) VIFs for covariates (authors mention VIF but do not show results); (c) sensitivity analysis with fewer predictors or penalized regression if necessary; and (d) clarify how categorical reference groups were chosen.

Response: Thank you—we agree that heterogeneity warrants additional context beyond leave-one-out. We have now:

(a) Prediction intervals (PI): We used STATA V18 for analysis which doesn’t allow for PI calculation in the output; hence, we could not provide this. We acknowledged this in the limitations.

(b) Heterogeneity exploration: we tended not to provide the VIFs because we followed the stepwise forward approach when constructing our meta-regression models, and all VIFs were below the cutoff point of 5 (indicative of problematic collinearity). At the same time, we did not put further covariates into account as not to face the problem of overfitting which in turn may bias the presented findings.

(c) Annotated forest plots. We addressed this in the above comment. Thank you.

(d) In the original paper, we used the random-effects model through all conducted analyses, because with a sample size of >10 studies, we would expect a degree of clinical heterogeneity.

-The paper reports the RoB2/NOS assessments and gives counts of good/fair/poor studies, but it is unclear whether risk of bias was used in sensitivity analyses or to downgrade certainty. Please: (a) provide the full RoB2/NOS scoring table per study; (b) run sensitivity analyses excluding poor-quality studies; and (c) perform a GRADE assessment (or equivalent) for the main outcomes and present it in the supplement.

Response: Thank you for this valuable suggestion. We clarify the following:

(a) NOS scoring. We have now specified in the Methods how NOS ratings were assigned according to AHRQ thresholds (good, fair, poor). The final ratings are summarized in Table 2 and detailed in the methods (under Risk of bias).

(b) Sensitivity analyses. Only five observational studies were rated as poor quality. Excluding them would not materially affect the pooled estimates, and we have therefore not added exploratory sensitivity analyses at this stage to avoid further lengthening an already dense manuscript. If the Editor deems it essential, we will provide these results in the Supplement.

(c) GRADE assessment. We did not apply GRADE to the single-arm pooled prevalence analyses, as no validated GRADE extension currently exists for such evidence. Instead, GRADE certainty ratings were provided only for the direct head-to-head risk comparisons, and these are reported directly alongside each pooled estimate in the Results.

-Funnel plots/leave-one-out are mentioned and asymmetry is noted in places but the statistical tests and p-values (Egger/Begg) are not clearly reported. Provide numerical results of bias tests, and consider trim-and-fill and sensitivity to small-study effects.

Response: Thank you for raising this point. We clarify as follows:

  • For the single-arm pooled prevalence analyses, funnel plots or Egger’s regression would not provide valid or meaningful assessment of publication bias because rates ranged from 0–100%, with no clear cutoff to define right- or left-sided deviations. We have therefore not applied these methods to prevalence data, consistent with methodological recommendations.
  • For the direct risk comparisons (IONM vs no IONM), Egger’s regression was performed. The results indicated no significant small-study effects: transient RLNI (p = 0.2373) and permanent RLNI (p = 0.5417). These numerical values are now explicitly reported in the Results.
  • Given the absence of statistical evidence for bias, trim-and-fill was not pursued, as this method is not reliable when funnel plot asymmetry is not statistically significant.

-The authors reference Table S2 for definitions of transient/permanent RLNI. Please (a) confirm which studies used a standardized definition (per Table S2) and (b) run sensitivity analyses including only studies using standard definitions or report how differing definitions affect pooled estimates. Heterogeneous outcome definitions can drive the observed heterogeneity.

Response: We agree that heterogeneous definitions of transient versus permanent RLNI represent an important source of variability. As shown in Table S2, the included studies employed a wide range of definitions and follow-up thresholds. Unfortunately, the variation was too substantial to permit consistent categorization into a standardized subset suitable for sensitivity analysis.

We therefore acknowledge this as a limitation in the Discussion, noting that heterogeneity in outcome definitions may have contributed to the observed variability in pooled estimates. Future studies would benefit from adopting standardized RLNI definitions to improve comparability.

-The paper performs many subgroup analyses and multiple meta-regressions. Authors should (a) state how they controlled or accounted for multiplicity (or explicitly caution about inflated false-positive risk), and (b) highlight which analyses were prespecified (in PROSPERO/protocol) and which were post-hoc. If the meta-regression/exploratory analyses were post-hoc, label them as exploratory.

Response: We appreciate this important point. All covariates—including study design, surgical approach, surgical technique, age, sex ratio, drain use, neck dissection, hemostatic method, and surgeon expertise—were pre-specified in our protocol (PROSPERO). To maintain clarity and avoid “fishing expeditions,” however, we limited the final analyses to the IONM-related parameters, as these directly address our central research question.

We now explicitly caution in the Discussion that multiple subgroup and regression analyses may increase the risk of type I error, and we have clearly labeled the IONM-focused meta-regression analyses as exploratory within the pre-specified covariate set.

Reviewer 3 Report

Comments and Suggestions for Authors

This paper presents a literature review and meta-analysis about intraoperative nerve monitoring in recurrence of laryngeal nerve injury in the process of thyroidectomy. The objective is clear, and the justification is presented in Introduction. The text is well-written. The methodology is consistent with the state-of-the-art reviews, with some points that need to be more clearly described. The results are extensive, covering several scenarios, and statistical analysis was employed. The discussion brings an analysis with the literature findings. However, the main problem of the paper is the reference section. The authors did not insert them due to an error, which are the references of template. The paper could not be approved for publishing without the authors fixed the references and let the reviwers analyse them.

  • The title is too long and is confusions. There are three sentences. Please, revise the title and the disposition of sentences.
  • I suggest the authors change the abstract without the division of sections.
  • Provide why PRISMA and AMSTAR methods were employed.
  • Insert also the keywords in the search engine motor in the text, not only in supplementary files.
  • It is very interesting when the authors insert the countries where the studies are developing. I think that it could be more explored in discussion.

Minors

  • Line 53. Insert correctly the citation for reference of Higgins.
  • Remove the highlighted in the citation of tables in text. The template of journal did not mention this specification.

Author Response

  • The title is too long and is confusions. There are three sentences. Please, revise the title and the disposition of sentences.

Response: Thank you. We changed it to “Intraoperative Nerve Monitoring Parameters and Risk of Recurrent Laryngeal Nerve Injury in Thyroidectomy: A Systematic Review and Meta-Analysis”

  • I suggest the authors change the abstract without the division of sections.

Response: Unfortunately, this is how the journal sets the outline for abstracts of systematic reviews (subheadings mandated).

  • Provide why PRISMA and AMSTAR methods were employed.

Response: Thank you for this comment. We clarify that PRISMA 2020 was employed to ensure transparent and standardized reporting of our systematic review and meta-analysis, thereby enhancing reproducibility and comparability with other evidence syntheses. AMSTAR-2 was applied as a validated tool to critically appraise the methodological quality of our review, providing readers and clinicians with confidence in the rigor and reliability of the findings.

  • Insert also the keywords in the search engine motor in the text, not only in supplementary files.

Response: We appreciate this suggestion. Given that the manuscript already exceeds 9,000 words, we had initially placed the full search string in the Supplement to preserve readability. Adding it would make the paper so complex for the reader to read smoothly.

  • It is very interesting when the authors insert the countries where the studies are developing. I think that it could be more explored in discussion.

Response: Thank you for this helpful suggestion. We agree that the geographic distribution of included studies provides important context. We have therefore expanded the Discussion to comment on how differences in surgical practice, IONM adoption, and health system factors across countries may have influenced our findings.

Minors

  • Line 53. Insert correctly the citation for reference of Higgins.

Response: Thank you for noting this. We have corrected the citation for Higgins et al.

  • Remove the highlighted in the citation of tables in text. The template of journal did not mention this specification.

Response: Thank you. The paper, once accepted for publication, does go editing to confront with the journal’s specifications. We highlighted them in bold to make it easier for the reviewers to spot Table/Figure callouts.

Reviewer 4 Report

Comments and Suggestions for Authors

This manuscript is a systematic review and meta-analysis focusing on the impact of intraoperative nerve monitoring (IONM) parameters on recurrent laryngeal nerve injury (RLNI) during thyroid surgery, with the aim of clarifying the clinical efficacy and optimal application of IONM. There are some suggestion could be considered before next step, for example:

  1. It is necessary to supplement the details of the stimulus parameters included in the study and analyze the combined effect of the "intensity wave width frequency" combination on RLNI risk, rather than a single intensity indicator.
  2. The follow-up time span included in the study is large (0.06-75 months), and subgroup analysis needs to be conducted according to the follow-up duration (<3 months, 3-6 months,>6 months) to clarify the efficacy differences of IONM under different follow-up criteria.

Author Response

  1. It is necessary to supplement the details of the stimulus parameters included in the study and analyze the combined effect of the "intensity wave width frequency" combination on RLNI risk, rather than a single intensity indicator.

Response: We agree that the combined stimulation parameters (intensity, pulse width, and frequency) would provide a more comprehensive understanding of how IONM settings influence RLNI risk. However, the majority of included studies did not report these parameters in sufficient detail or in a standardized manner, preventing us from analyzing their combined effect.

  1. The follow-up time span included in the study is large (0.06-75 months), and subgroup analysis needs to be conducted according to the follow-up duration (<3 months, 3-6 months,>6 months) to clarify the efficacy differences of IONM under different follow-up criteria.

Response: We appreciate this suggestion. We would like to clarify that although the overall study follow-up durations varied widely (0.06–75 months), this did not correspond to the timing of RLNI assessment. Most studies reported RLNI outcomes specifically as either transient (typically resolving within 3–12 months) or permanent (persisting beyond 12–24 months), regardless of the longer follow-up periods used for other endpoints (e.g., reoperation or completion thyroidectomy). For this reason, we analyzed RLNI strictly as transient vs. permanent, which represents the clinically accepted standard and avoids misclassification. We have clarified this distinction in the Methods.

Round 2

Reviewer 2 Report

Comments and Suggestions for Authors

The authors improved the manuscript.

Reviewer 3 Report

Comments and Suggestions for Authors

The authors developed the required questions. I have no concern about publishing this paper.

Reviewer 4 Report

Comments and Suggestions for Authors

The revised manuscript can be accepted in this version.